# Impact of the First Wave of the COVID-19 Pandemic on HIV/AIDS Programming in Kenya: Evidence from Kibera Informal Settlement and COVID-19 Hotspot Counties

**DOI:** 10.3390/ijerph18116009

**Published:** 2021-06-03

**Authors:** Samuel Muhula, Yvonne Opanga, Violet Oramisi, Catherine Ngugi, Caroline Ngunu, Jane Carter, Enock Marita, Joachim Osur, Peter Memiah

**Affiliations:** 1Monitoring Evaluation and Research Unit, Amref Health Africa in Kenya, 30125-00100 Nairobi, Kenya; samuel.muhula@Amref.org (S.M.); enock.marita@amref.org (E.M.); 2Department of Strategic Health Programs, Division of National AIDS and STIs Control Programme, 19361-00202 Nairobi, Kenya; voramisi@nascop.or.ke (V.O.); cngugi@nascop.or.ke (C.N.); 3Health Directorate, Nairobi Metropolitan Services, 49130 Nairobi, Kenya; ngunucarol@yahoo.com; 4Clinical and Diagnostics Programme, Amref Health Africa Headquarters, 27691-00506 Nairobi, Kenya; jane.carter@amref.org; 5Office of the Vice Chancellor, Amref International University, 27691-00506 Nairobi, Kenya; joachim.osur@amref.org; 6Division of Epidemiology and Prevention: Institute of Human Virology, University of Maryland Baltimore, Baltimore, MD 21201, USA; pmemiah@ihv.umaryland.edu

**Keywords:** COVID-19, HIV/AIDS, antiretroviral therapy, food security, pre-exposure prophylaxis, lockdown measures, personal protective equipment

## Abstract

The study sought to determine the impact of COVID-19 on HIV/AIDS programming in the Kibera informal settlement and COVID-19 hotspot counties during the first wave of the pandemic. The study was conducted in two phases. The first phase entailed the analysis of HIV care and treatment secondary data (2018–2020) from the Kenya Health Information System. In the second phase, a prospective cohort study was conducted among people living with HIV in the Kibera informal settlement. A total of 176 participants aged 18 years and above accessing HIV services at selected healthcare facilities in Kibera were randomly sampled from facility electronic medical records and followed up for three months. Socio-demographics and contact details were abstracted from the records and telephone interviews were conducted with consenting participants. Results from the retrospective review of HIV program data indicated a 56% (*p* < 0.000, 95% CI: 31.3%–62.8%) reduction in uptake of HIV services. Clients starting antiretroviral therapy (ART) reduced significantly by 48% (*p* < 0.001, 95% CI: 35.4%–77%) in hotspot counties. However, pre-exposure prophylaxis uptake increased significantly by 24% (*p* < 0.019, 95% CI: 4%–49%). In Kibera, 14% reported missing medications at the onset of the COVID-19 pandemic because of lack of food (38%) and government measures (11%), which affected ART access; 11% did not access health facilities due to fear of contracting COVID-19, government regulations and lack of personal protective equipment. Socioeconomic factors, food insecurity and government measures affected uptake of HIV/AIDS services; hence, the need for scaling up measures to increase access to HIV/AIDS services during the onset of pandemics.

## 1. Introduction

Kenya has witnessed tremendous progress over the last decade in reducing the annual number of new HIV infections and AIDS-related deaths, resulting in improved quality of life for people living with HIV (PLHIV) and reduced risk of infection. According to the Kenya population-based HIV impact assessment (KENPHIA) 2018 preliminary report, 79.5% of PLHIV in the country are aware of their HIV status, 96% of people who know their HIV status are on life-saving treatment and 90.6% of those had achieved viral load suppression [1]. In addition, Kenya has been a prevention success case study in the region, not only being the first country to approve the use of pre-exposure prophylaxis (PrEP) [2], but also leading in the uptake of preventive measures such as voluntary medical male circumcision [3]. As a result, new HIV infections have reduced drastically in recent years [4].

Kenya reported its first case of COVID-19 on 13 March 2020 [5]. As of 23 February 2021, there were 104,500 cases, 1837 deaths and 85,665 recoveries reported in the country [5]. High numbers of COVID-19 cases in Kenya were first reported in what was defined as “hotspot” counties [6] because of their proximity to Nairobi (the first county to receive an imported case) or their proximity to international borders, which led to border contamination from neighboring countries. These counties, Nairobi, Mombasa, Nakuru, Kajiado, Busia, Kiambu and Machakos, reported high numbers of cases of COVID-19 in Kenya in the early months after the first case was reported. By the end of December 2020, Nairobi had registered over 40,000 confirmed cases, which represented over 40% of the total cases in Kenya [5]. Mombasa, Kenya’s oldest and second largest city, reported over 8800 cases, while Kiambu and Machakos, which are within the Nairobi metropolitan region, reported over 6300 and 2400 confirmed cases, respectively [5]. Kajiado and Busia Counties border Tanzania and Uganda, respectively, where the Government of Kenya was challenged with the management of cross-border infections without interfering with the free movement of goods across the East African countries. The two counties had recorded over 3100 and 2700 cases respectively, by the end of December 2020; Nakuru County reported over 4700 COVID-19 cases [5,7]. By the end of the study period in February 2021, the world had recorded about 116 million COVID-19 cases and 2,589,548 deaths [8].

People living in resource-limited settings such as informal settlements, and women and children, are particularly vulnerable during pandemics and experience a relatively higher disease burden associated with the pandemic [9]. The largest informal settlement in Kenya is the Kibera slums. The slum is located 5 kilometers south west of Nairobi’s Central Business District and comprises 14 villages (Kianda, Olympic, Soweto West, Gatwikira, Karanja, Kisumu Ndogo, Raila, Makina, Kambi Muru, Mashimoni, Lindi, Laini Saba, Silanga and Soweto East) with a population of 185,777 individuals [10]. The area covers approximately 12.1 km^2^, is densely populated with 15,311 persons per square kilometer and has a high burden of infectious diseases [11,12,13]. Kibera residents suffer from a host of challenges such as lack of clean reliable water, poor housing, inadequate health services and lack of solid waste management facilities [14]. These conditions make it difficult for the residents of Kibera to practice physical distancing measures, which are critical strategies to slowing the spread of COVID-19 [15,16]. Further, it is not known how the first wave of COVID-19 affected PLHIV’s care-seeking behavior amid government-enforced movement restrictions, particularly in Nairobi and Mombasa Counties. Therefore, in the Kenyan context, the effect of COVID-19 on HIV/AIDS care and treatment programs remains unknown. Additionally, during the first wave of the pandemic, there were few studies to establish any increased risk of infection, increased severity of illness among PLHIV, interruption in access to HIV medication and services or any correlation between COVID-19 and HIV/AIDS. Studies have shown that people with pre-existing conditions who contract COVID-19 are at higher risk of disease progression including death [17,18] though little is known about the relationship between COVID-19 and HIV/AIDS.

The primary objective of this study was therefore to determine the impact of COVID-19 on HIV/AIDS programming during the first wave of the pandemic by gathering evidence from Kibera informal settlement and COVID-19 hotspot counties in Kenya. The specific study objectives were (1) to determine the impact of COVID-19 on healthcare-seeking behavior of PLHIV in the Kibera informal settlement and (2) to determine the impact of COVID-19 on PrEP uptake and HIV testing and treatment services in COVID-19 hotspot counties in Kenya. This study aimed to provide credible and clear answers to these knowledge gaps during the first wave of the pandemic, with the hope that the findings would help in decision making by the Ministry of Health COVID-19 Task Force, county governments and many other stakeholders working in HIV/AIDS care and treatment in Kenya.

## 2. Materials and Methods

Data were collected in two phases to answer the objectives. The first phase entailed the analysis of HIV care and treatment secondary data from the Kenya Health Information System (KHIS) between 2018 and 2020 in the seven hotspot counties. Data were abstracted for the following variables: (1) number of reported cases utilizing HIV testing and counseling services, (2) number of cases utilizing and initiated on PrEP and (3) number of cases started on ART. Data were collected from January 2018 to June 2020 to compare the variables before and after COVID-19 in the selected counties.

The second phase entailed a prospective cohort study of PLHIVs from the Kibera informal settlement in Nairobi, who were followed up monthly for three consecutive months during the first wave of the pandemic in Kenya. Data for the prospective cohort component of the study were collected from seven healthcare facilities supported by Amref Health Africa under the US Centers for Disease Control and Prevention-funded Kibera Reach 90 project [19]. These were: KEMRI Mimosa, Amref Kibera Community Health Center, Kibera South (MSF Belgium) Heath Center, Langata Health Center, Silanga (MSF Belgium) Dispensary, Tabitha Medical Clinic and Ushirika Medical Clinic. These health facilities covered the 10 sublocations in the Kibera informal settlement and therefore represented the Kibera informal settlement population. The seven health facilities had a total of 10,480 patients on HIV care and treatment at the start of this study in March 2020.

The study population comprised of 176 PLHIVs aged 18 years and above accessing HIV care and treatment in the seven health facilities. The sample size included an additional 5% to cater for loss to follow up. The sampling frame was the total number of 10,480 HIV/AIDS-positive clients who were receiving treatment in the seven healthcare facilities (Table 1) as of March 2020. This number was confirmed from the hospital records at the time of the study. We used the Cochran sample size calculation formula to estimate the proportion of HIV-positive patients to be interviewed. Using the formula, p is the proportion of the population in Kibera that is HIV positive, which equals 12.6%; Z^2^ was the desired confidence level, 1 – α, which equals 95% = 1.96; e is the desired level of precision at 0.05. The target sample size was 176 PLHIVs, distributed proportionate to the number of PLHIVs receiving care and treatment at the seven health facilities (Table 1).

The team of research assistants sought permission from the facility in charge and introduced the study in addition to presenting approval letters from the Nairobi County authorities, the National Commission for Science Technology and Innovation (NACOSTI) and Amref’s Ethics and Scientific Review Committee (ESRC). Each research assistant presented a list of the variables required to the data officers at the facilities for abstraction from the facility electronic medical records systems (EMR). The list of all participants from the seven facilities was abstracted from the EMRs at the health facilities. The list was provided to the principal investigator who then randomized the study participants, produced print outs and shared with the research assistants for scheduling telephone interviews. The participants were informed about the study by the healthcare workers, and the research assistants then called the participants at least thrice in a span of one hour to obtain their consent to proceed. Those who accepted to be interviewed were taken through the consenting process before interviews were conducted. Those who did not respond or declined to be interviewed were replaced by drawing another random sample from the database. During the interviews, the research assistants called the participants at least twice in the span of one hour and sent text messages to the participants who did not respond. Non-responding participants were replaced after 24 h by selecting the next participant in the database. Research assistants introduced the study to those who responded and scheduled an appropriate time for interviews. The telephone interviews were conducted within the premises of the health facilities in a quiet and confidential room, or a secure room within the Amref Health Africa in Kenya offices in instances where the health facilities did not have rooms suitable for interviews.

Data were collected in July, August and October 2020 through telephone-based interviews, where sampled respondents were asked the survey questions. The enumerators entered the responses directly into open data kit (ODK) on handheld tablets. An average of 35 min was used to complete the interview for each participant. No incentives were given to participants, but they were invited to select a convenient time for the interviews to take place. To adhere to government regulations on controlling the spread of COVID-19, all research assistants were provided with adequate face masks and hand sanitizers during the data collection period and asked to observe a safe physical distance. Upon entry into the health facilities, the research assistants were accompanied by at least one study investigator. All the data were kept under password-protected files and encrypted for safety with only the study team having access to the records. The collected data were downloaded into Microsoft Excel, further cleaned and imported into STATA version 15 for analysis involving generating descriptive statistics such as means, medians, frequencies and cross-tabulations. Regression analysis was conducted to assess the relationship between COVID-19 (outcome variable) and HIV risk factors.

## 3. Results

### 3.1. Demographic and Clinical Characteristics

A total of 176 (104%), 150 (89%) and 151 (89%) respondents were interviewed in rounds one, two and three, respectively. During the study, there was an 11% loss to follow up between the first and final rounds of data collection. Reasons for loss to follow up in rounds 2 and 3 were as follows: (1) participants stated they preferred face-to-face interviews, (2) no response to telephone calls after two attempts, (3) participants complained of being disturbed, (4) participants were working and agreed to a later call but never responded, (5) participants stated they were not interested in the study and (6) someone else answered the telephone but did not relay the message. The study sample had a higher proportion of female respondents (65%); the majority (35%) of respondents were in the age group 35–44 years. The youngest respondent was aged 18 years and the oldest was 70 years. About 75% of respondents did not complete primary education, while only 1% had completed university education. The majority (38%) were casual laborers, while 23% were unemployed. Almost all (98%) respondents were Christians. The average household size in Kibera was four members. Almost all (93%) respondents were first diagnosed with HIV more than one year before data collection. The respondents’ viral load levels remained stable throughout the four month period of data collection. About 25% of respondents were TB co-infected. Table 2 shows details of the demographic and clinical characteristics of the study participants.

### 3.2. Effects of COVID-19 on Healthcare-Seeking Behavior among PLHIV in Kibera Informal Settlement

The proportion of respondents who consistently missed medication over the four month period reduced from 14% (*n* = 24, 95% confidence interval (CI): 0.1%–27.9%) in July 2020 to 7% (*n* = 10, 95% CI: −8.8–22.8%) in August and 5% (*n* = 8, 95% CI: −10.1%–20.1%) in October 2020, although this was not significantly different as demonstrated by the overlapping confidence intervals. The main reasons for missing medication were forgetting to take medicines followed by running out of medication, as shown in Figure 1. More than one-third (38%) of respondents also reported that they did not have food in the house in October 2020 and therefore could not take the medication.

The proportion of respondents who tested for COVID-19 reduced from 12% (*n* = 21, 95% CI: −1.9%–25.9%) in July 2020 to 9% (*n* = 13, 95% CI: −6.6%–24.6%) in August 2020 and 4% (*n* = 6, 95% CI: −11.7%–19.7%) in October 2020. All respondents previous heard of the coronavirus disease.

Across the four months, most respondents (July—46%, August—42%, October—39%) visited their health facilities more than one month prior to the survey start date for medication refills. This was expected given that, on average, 97% of the PLHIVs were stable (viral load of <1000 copies/mL) on treatment.

We sought to determine if the COVID-19 screening done at the entrance to health facilities would affect patients’ routine visits to the health facilities. Almost all (98%) the respondents mentioned that this did not in any way affect their visits to the health facilities.

In July 2020, 11% of the respondents indicated that the measures put in place by the government to control the spread of coronavirus affected their access to ART. The issues mentioned included patients not able to travel because of lockdown measures, some health facilities being closed and turned into COVID-19 centers or closed for fumigation, and shortage of Cotrimoxazole (Septrin). This proportion of patients reduced to 4% in August and 2% in October since most of the government restrictions were lifted by these dates.

Similarly, 11% of the respondents in July, 3% in August and 1% in October 2020 reported challenges in accessing the health facilities as a result of fear of contracting COVID-19 and government lockdown regulations, and some of the respondents did not have personal protective equipment such as face masks and lacked the fare to reach the health facilities.

About 23% of the respondents in July 2020 indicated they had fears and concerns about seeking healthcare services in a health facility. That proportion reduced to 13% in August and to 5% in October 2020. The main fears mentioned were the fear of contracting coronavirus, fear of being tested for coronavirus and fear of unavailability of drugs at the health facilities. Figure 2 shows the main fears and concerns about seeking healthcare services in health facilities.

About 17% of the respondents in July, 20% in August and 14% in October 2020 reported having experienced at least one of the COVID-19 symptoms. However, there was no exponential increase in any of these symptoms throughout the four month period. All (100%) respondents knew of the coronavirus disease before this study and about two-thirds (62%) sought medical care whenever they experienced COVID-19 symptoms.

To gauge the impact of COVID-19 on daily life, we asked the respondents how much their lives had changed since the outbreak of the pandemic. In July and August, 96% and 95% of the respondents, respectively, indicated that their lives had changed in one way or another. This proportion reduced to 70% in October 2020. The top three ways in which the respondent’s lives had changed were reduced income from employment or work, increased hunger and increased cost of living.

We also looked at how COVID-19 affected access to sufficient, safe and nutritious food that meets dietary needs, and the food preferences for members of households in Kibera informal settlement in the four weeks before data collection. About 19% of the respondents in July, 21% in August and 23% in October indicated that they went at least one whole day and night without eating anything because there was not enough food. Some (51% in July, 51% in August and 42% in October) indicated that they did not have any kind of food to eat in their households because of lack of resources to buy food, while 82% of the respondents in July, 82% in August and 74% in October worried that their households would not have enough food to eat. Figure 3 shows details of the issues around household food insecurity.

### 3.3. Impact of COVID-19 on HIV Testing and Counseling Services, Pre-Exposure Prophylaxis (PrEP) Uptake and HIV Treatment Services

We assessed the impact of COVID-19 on PrEP, HIV testing and counseling and HIV treatment services uptake in the seven COVID-19 hotspot counties, two years before and four months after the COVID-19 outbreak in Kenya.

HIV testing and counseling uptake in the seven COVID-19 hotspot counties was significantly affected by the onset of the pandemic. From a Poisson regression analysis, we found that service uptake reduced by 56% (*p* < 0.000, 95% CI: 31.3–62.8%) with the onset of the COVID-19 pandemic, holding other variables constant, as presented in Table 3.

Pre-exposure prophylaxis uptake among discordant couples, the general population, men having sex with men, female sex workers and people with disabilities significantly increased by 24% (*p* < 0.019, 95% CI: 4–49%) from the onset of the pandemic, holding other variables constant, as shown in Table 4.

Clients starting ART significantly reduced by 48% (*p* < 0.001, 95% CI: 35.4%–77%) from the onset of the pandemic, holding other variables constant as indicated in Table 5

## 4. Discussion

In this study, approximately 97% of the participants achieved viral suppression, and this was higher than the national level, which now stands at 90.6% [20] A similar trend was also observed in the hotspot counties. Data from the Kenya Health Information System indicated viral suppression in hotspot counties was lower than that reported among participants in Nairobi (before 93.6%; during 93.4%), Mombasa (before 94.3%; during 92.4%), Machakos (before 82.4%; during 92.6%), Busia (before 95.2%; during 93.5%), Kiambu (before 93.3%; during 93.8%), Nakuru (before 94.3%; during 93.8%) and Kajiado (before 91.4%; during 90.7%). In the last decade, efforts towards scaling up HIV care, treatment and prevention in Kenya have been successful. Among adults aged 15–64 years, HIV incidence is now at 0.14% (36,000 cases), 79.5% who tested positive now know their status, 96% who know their status are on ART and 90.6% have achieved viral suppression [20]. The COVID-19 pandemic, however, posed a threat to these achievements since HIV prevention, care and treatment programs have been affected by the stringent government measures to constrain the spread of COVID-19.

This study indicated a 56% reduction in the uptake of HIV prevention, care and treatment services in hotspot counties. In addition, clients starting ART significantly reduced by 48% in the hotspot counties. However, pre-exposure prophylaxis uptake significantly increased by 24%. During the first months of the pandemic, the government prioritized interventions on controlling the spread of the SARS-CoV-2 virus, and less effort was put into ensuring continuity of other healthcare services, including delivery of HIV services [21]. Other challenges that disrupted the uptake of HIV services included the following: stringent government measures that limited movement, thus reducing access to healthcare services; the closure of healthcare facilities either for fumigation or for use as COVID-19 diagnosis and treatment centers; the limited availability of personal protective equipment for participants, which was a requirement to access facilities; few healthcare workers turning up to work for fear of exposure; and shortages of essential drugs due to border closures [22]. Additionally, some participants moved from urban centers to rural areas to flee from COVID-19 [23] but were later affected by the lockdown, making them unable to access their source of ARVs. The World Health Organization later shared guidelines [24] on the continuity of essential services, which were adopted by the Ministry of Health [25]. Therefore, at the onset of the pandemic, the government through the Ministry of Health needed to develop a framework to ensure continuous provision of HIV/AIDS services as recommended by USAID [22].

After the first case of COVID-19 was reported in Kenya in March 2020, the government through the Ministry of Health set up a National Emergency Response Committee to help mitigate the effects of the pandemic [26]. Some of the directives from the committee included closure of schools, suspension of international flights, mandatory quarantine of people flying in from other countries, enactment of a nationwide curfew from 7 p.m. to 5 a.m., cessation of movement in informal settlements, observance of physical distancing (1.5 m), proper wearing of personal protective equipment, regular handwashing with soap and water, screening points and handwashing stations at every healthcare facility and limited patient visitation at all public and private hospitals [16,26]. At the facility level, efforts were geared towards infection prevention and control, healthcare worker safety, reduced crowding and management of COVID-19 patients [27]. Travel disruptions, lockdowns, limitation of movement and curfew measures interrupted access to healthcare facilities making it harder for participants to obtain or refill medications [24]. Additionally, a major reason for missing ART medication was that the majority of participants forgot because of increased pressures from food insecurity, HIV status and loss of livelihoods, among other factors that were magnified by the pandemic. The proportion of participants missing drugs, however, reduced from 7% in August to 5% in October 2020. This could be explained by the progressive reopening of the country by late July 2020. Movement restrictions in Nairobi and Mombasa were lifted and the night curfew moved from 7 p.m.–4 a.m. to 9 p.m.–5 a.m. [16]. Therefore, there is a need for the government to intervene in shielding PLHIV from such stresses during a pandemic to ensure adherence to ART.

Shortages of some medications such as Cotrimoxazole (Septrin) in healthcare facilities was also a contributing factor to missing medication. HIV service delivery was interrupted during the pandemic including the drug supply chain. A failure of suppliers to deliver ARVs to facilities on time was due to lockdown challenges, shutting down borders and interruptions in land and air transport together with limited access to health services within countries [28]. The guidelines on access to essential medicines developed by the World Health Organization recommended that countries adopt multi-month dispensing policies, use of digital platforms and telephone calls and home visits to ensure ARVs are prescribed for longer periods, such as 6 months [24,28]. Following this directive, the multi-month dispensing of medicines and use of m-health applications such as *USHAURI* (an SMS-based patient reminder system developed to enhance client retention in care and improve adherence to treatment) to communicate to clients on service availability and appointment reminders, were adopted by the Ministry of Health [25].

Food insecurity was a predominant factor that affected the uptake of medications. More than one-third (38%) of participants reported insufficient food in the household. The strict control measures in response to the COVID-19 pandemic negatively affected the food and economic security of informal settlement residents [16]. A study on the impact of COVID-19 control measures on social contacts and transmission in Kenya informal settlements confirmed that the greatest unmet need during the pandemic was food, with over 70% reporting eating less or skipping meals due to lack of funds to access food [16]. The food insecurity trend was observed to increase rather than decrease during the study period. For instance, about 19% of the respondents in July, 21% in August and 23% in October indicated that they went at least one whole day and night without eating anything because there was not enough food. Informal settlements are predominantly populated by residents with poor socioeconomic status who rely mostly on the informal sector for their sources of income and livelihoods [29]. These populations are often vulnerable to food insecurity, and this worsened due to disruptions in livelihoods, closure of informal sectors leading to reduced income, loss of jobs and limited movement, which prevented the participants from conducting daily activities. Although some participants reported receiving food aid, this did not resolve the issue. There is therefore a need for the government and development partners to develop mechanisms to cushion food security through social protective measures amid the stringent control measures put in place to manage a pandemic.

The study indicated only one positive case of COVID-19 among those who were tested. Although it was predicted that Africa’s informal settlements would be the hardest hit by high levels of infection due to poor conditions of hygiene and inability to practice social distancing [23], this appeared not to be the case. This could be because many participants fled to the countryside, non-governmental organizations initiated community-driven water hygiene and sanitation interventions, such as providing free drinking water and handwashing stations, and the public was sensitized on a daily basis about COVID-19 [23]. Additionally proactive members of the settlement made efforts to ensure government measures were followed, including handwashing, use of PPEs and limited movement [23]. From this study, it is evident that participants were more affected by social factors, such as food insecurity and changes in livelihoods and socioeconomic status, rather than by COVID-19 infection.

This study is limited to findings on the impact of the first wave of COVID-19 on HIV/AIDS programming and on livelihoods of PLHIV. This situation may have changed during the second and third waves. We therefore recommend further research to establish whether there were any significant changes in the management of programs and livelihoods of PLHIV during subsequent waves of the COVID-19 pandemic.

## 5. Conclusions

The lives of HIV patients have been negatively impacted in several ways since the start of the pandemic. In one informal settlement in Kenya, patients experienced challenges accessing health facilities and ART, especially in the month of July 2020, because of compounding issues such as fear of contracting COVID-19, closure of health facilities, government lockdown measures and lack of personal protective equipment. HIV patients were also at high risk of food insecurity and malnutrition because of the inability to fend for themselves and their household members. The situation, however, was improved as of October 2020. HIV testing services and ART initiation were also reduced since the onset of the COVID-19 pandemic. Further research is required to understand the effects of COVID-19 co-infection of PLHIVs.

## Figures and Tables

**Figure 1 ijerph-18-06009-f001:**
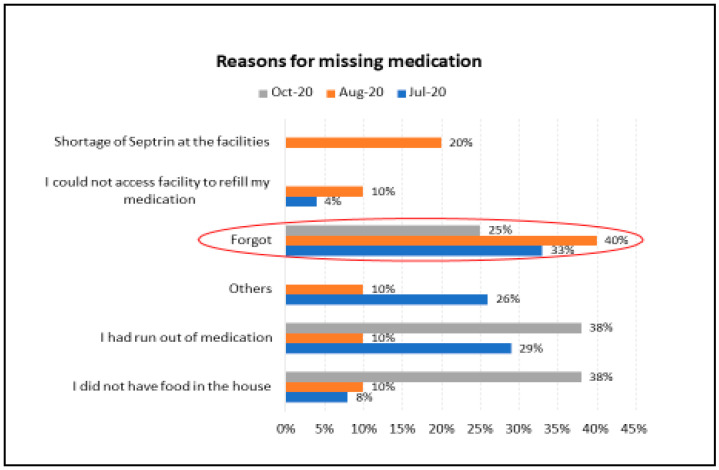
Reasons for missing ART medication.

**Figure 2 ijerph-18-06009-f002:**
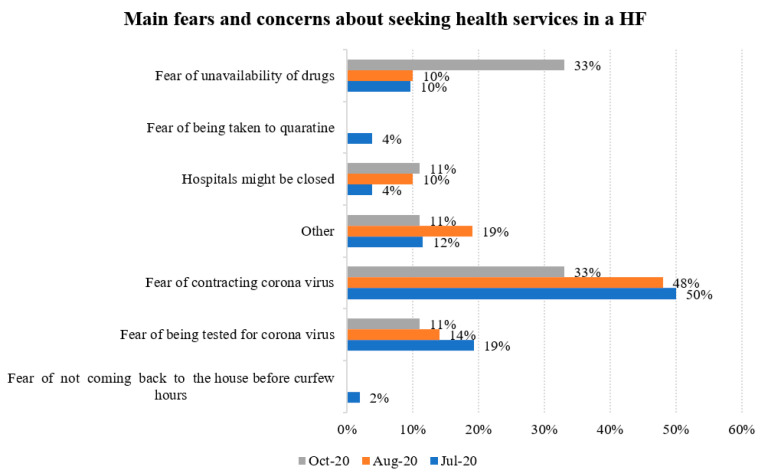
Main fears and concerns about seeking health services in health facilities.

**Figure 3 ijerph-18-06009-f003:**
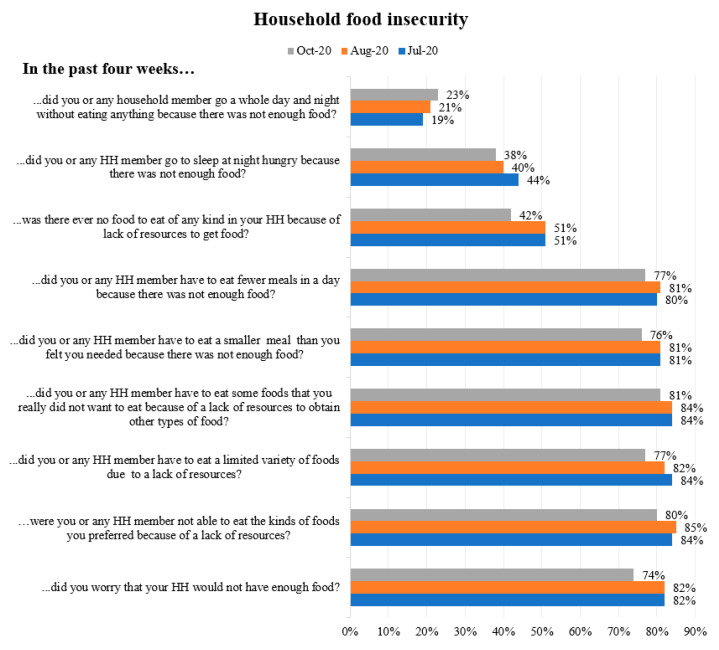
Household food insecurity.

**Table 1 ijerph-18-06009-t001:** Sample size distribution by site.

Health Facilities	Total Number of PLHIV on Care and Treatment	Selection Probability	Sampled Clients
Kemri Mimosa	801	0.08	13
Kibera Community Health Center—Amref	2395	0.23	39
Kibera South (MSF Belgium) Health Center	3551	0.34	58
Langata Health Center	885	0.08	14
Silanga (MSF Belgium) Dispensary	1370	0.13	22
Tabitha Medical Clinic	715	0.07	12
Ushirika Medical Clinic	763	0.07	12
Total	10,480	1	170

**Table 2 ijerph-18-06009-t002:** Demographic and clinical characteristics.

Variable	*n* (%)
Gender	
Male	61 (35%)
Female	115 (65%)
Age groups	
18–24	3 (2%)
25–34	32 (18%)
35–44	62 (35%)
45–54	54 (31%)
55+	25 (14%)
Level of education completed	
None	54 (31%)
Primary	77 (44%)
Secondary	25 (14%)
University	2 (1%)
Occupation	
Salaried employment	25 (14%)
Self-employment/business person	39 (22%)
Casual labor	67 (38%)
Unemployed	40 (23%)
Others	4 (2%)
Marital status	
Single/Never married	21 (12%)
Married	102 (58%)
Divorced/Separated	27 (15%)
Widowed	26 (15%)
Household size	
Respondent only	11 (6%)
Two	19 (11%)
Three	23 (13%)
Four	34 (19%)
Five	35 (20%)
Six	26 (15%)
Seven+	28 (16%)
Religion	
Christian	172 (98%)
Muslim	4 (2%)
1st HIV diagnosis	
More than 1 year ago	164 (93%)
Past 1 year	10 (6%)
Past 6 months	1 (1%)
Past 1 month	1 (1%)
Viral load (copies/mL)	
<1000	170 (97%)
>1000	6 (3%)
HIV/TB co-infection	
Yes	44 (25%)

**Table 3 ijerph-18-06009-t003:** Uptake of HIV testing and counseling before and during COVID-19.

HIV Testing	Incidence Rate Ratio	Robust Standard Error	z	*p* > ǀzǀ	95%Confidence Interval
Month	1.054963	0.027901	2.02	0.043	1.001671–1.11109
Year	0.8261786	0.2183384	−0.72	0.470	0.4921785–1.386836
Period	0.4436609	0.0787972	−4.58	0.000	0.3132362–0.6283916
Constant	1.9e + 172	1.0e + 175	0.74	0.457	3.1e–282

**Table 4 ijerph-18-06009-t004:** Uptake of pre-exposure prophylaxis before and during the COVID-19 pandemic.

PREP	Incidence Rate Ratio	Robust Standard Error	z	*p* > ǀzǀ	95%Confidence Interval
Month	0.9739061	0.0091511	−2.81	0.005	0.9561345–0.0020081
Year	1.241527	0.1143043	2.35	0.019	1.036545–1.487045
Period	1.542447	0.1897687	3.52	0.000	1.211954–1.963063
Constant	0	0	−3.50	0.000	0–2.2e–166

**Table 5 ijerph-18-06009-t005:** Clients starting ART before and during the COVID-19 pandemic.

Starting ART	Incidence Rate Ratio	Robust Standard Error	z	*p* > ǀzǀ	95%Confidence Interval
Month	1.035541	0.0168434	2.15	0.032	1.003049–1.069085
Year	0.5237438	0.1044745	−3.24	0.001	0.3542618–0.7743075
Period	1.797552	0.4962637	2.12	0.034	1.046367–3.088013
Constant	0	0	−2.11	0.035	0–6.99e–37

## Data Availability

All data are available upon request.

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
