# Peer review of "Impact of the First Wave of the COVID-19 Pandemic on HIV/AIDS Programming in Kenya: Evidence from Kibera Informal Settlement and COVID-19 Hotspot Counties"

_ijerph, 2021, doi:10.3390/ijerph18116009_

Round 1

Reviewer 1 Report

This paper addresses a central question about continuity of care during an epidemic. The article is also interesting because it deals with the beginning of the pandemic. It would be interesting to see how the impact of the pandemic evolved in the second wave.

If the method used to sample from an electronic medical record and the list of HIV patients, there are elements that seem problematic to me.  The total number of patients retained for the survey is 170 in the results section, 176 in the summary. Beyond this error in the figures and even if the authors rely on the Cochran sample size calculation method, I am puzzled by the size of the total sample when more than 10,000 patients are registered in the database. This sample size decreases between rounds to reach 150 patients in the last round, which is very small for a 7 hotspot reading (hotspot not defined by the authors).

Moreover, we have little information on the questionnaire, and the vagueness is created since the authors speak of interviews which evokes more a qualitative approach. But the quantitative results leave little doubt. 

The discussion elements are interesting as well as the subject of the article but the weakness of the sample and the interpretation by categories and hotpsot counties seems to me really weak and does not constitute a solid basis for the publication of this article.

Author Response

Dear Reviewer 1,

Many thanks for the comments that have helped improve our manuscript. Find below the comments. Also please see attachment of point by point rresponse letter and questionnaire used for the prospective cohort phase.

Reviewer comment

Responses

Line number

1

This paper addresses a central question about continuity of care during an epidemic. The article is also interesting because it deals with the beginning of the pandemic. It would be interesting to see how the impact of the pandemic evolved in the second wave.

We appreciate the reviewer’s comment; however, the study was conducted during the first wave of the pandemic in Kent and the results included are limited to the beginning of the pandemic. This has been reflected in the title, abstract and also included in the discussion section as a limitation of the study and aspect noted for future research.

Line 2, 13, 381-385

2

If the method used to sample from an electronic medical record and the list of HIV patients, there are elements that seem problematic to me.  The total number of patients retained for the survey is 170 in the results section, 176 in the summary. Beyond this error in the figures and even if the authors rely on the Cochran sample size calculation method, I am puzzled by the size of the total sample when more than 10,000 patients are registered in the database. This sample size decreases between rounds to reach 150 patients in the last round, which is very small for a 7 hotspot reading (hotspot not defined by the authors).

The sample size of 170 PLHIV was properly calculated using the Cochrane sample size calculation formula. We however collected data from 176 PLHIVs.  The methodology has been revised to clarify the issues raised. Phase one entailed secondary analysis of HIV/AIDS indicators on care and treatment in hotspot counties from Kenya Health information system. The second phase entailed primary data from respondents in Kibera Informal settlements.

In methods line 101-132 and in the results section line 179-185

3

Moreover, we have little information on the questionnaire, and the vagueness is created since the authors speak of interviews which evokes more a qualitative approach. But the quantitative results leave little doubt. 

The questionnaire has been attached for reference and can be added as supplemental file as guided

4

The discussion elements are interesting as well as the subject of the article but the weakness of the sample and the interpretation by categories and hotpsot counties seems to me really weak and does not constitute a solid basis for the publication of this article.

The methodology section has been revised to clarify the issues raised. We are happy to provide further clarifications in case what we have shared is not adequate

In methods line 101-132

Reviewer 2 Report

The article deals with an important issue, it is well funded and well written.

The relation of HIV vulnerability, COVID-19 risk and food insecurity is well presented and discussed in the paper. Conclusions are fine and highlight the importance of introducing measures in low income communities to prevent famine.

The only point I consider could be further analysed and discussed is if there were changes in sexual behaviour among the studied persons, during the pandemic studied period, as social distance was recommended as a preventive measure for COVID-19 by health authorities.

Author Response

Dear Reviewer,

We really appreciate your comments because they have helped improve our manuscript. Please see attached comments. Thank you

Kind Regards,

Yvonne Opanga

Reviewer 3 Report

  1. From reading the abstract, it is not immediately clear if there are two data sources (a prospective cohort study and a retrospective review). If there are two data sources, the authors should consider saying that a retrospective review was *also* conducted. 
  2. Line 19- The acronym ART should be spelled out the first time it is used.
  3. Lines 50-53 – The authors make a compelling argument that currently, there is a lack of data on the effects of COVID-19 co-infection in PLHIV. Their statement “At present, there is no evidence to suggest whether there is an increased risk of infection or increased severity of illness among PLHIV, interruption in access to HIV medication and services, or any correlation between COVID-19 and HIV/AIDS” seems to be a sweeping statement or generalization. The authors can consider either providing a citation to support this statement or tempering this statement to say that this is according to the authors’ knowledge, or that there are few studies that study this phenomenon. 
  4. Regarding data collection, how long did the telephone-based interviews last? Was Also, were incentives provided for participants to participate?  In the case where there were no monetary incentives, were there other considerations to acknowledge people’s time?
  5. Being that data was collected at three points (July, August, and October 2020), was this the same survey administered to three different groups of people? Was the retrospective review used to supplement the survey?  When is the retrospective review factored into the analysis?  The methodology is not quite clear.
  6. Overall, this is a pertinent, important, and timely topic.  The methods of the paper are a bit unclear.  Attending to these clarifying questions may help to improve the paper.

Author Response

Dear Reviewer,

Many thanks for the valuable comments shared. They have helped improve our manuscript. Please find attached the point by point response. Thank you

No.

Comment

Response

Line Number

1

From reading the abstract, it is not immediately clear if there are two data sources (a prospective cohort study and a retrospective review). If there are two data sources, the authors should consider saying that a retrospective review was *also* conducted. 

The abstract has been revised to reflect the suggestion made

Line 12-20

2

Line 19- The acronym ART should be spelled out the first time it is used.

The acronym has been spelt out in the abstract and when it is first used in the narrative

Line 22

3

Lines 50-53 – The authors make a compelling argument that currently, there is a lack of data on the effects of COVID-19 co-infection in PLHIV. Their statement “At present, there is no evidence to suggest whether there is an increased risk of infection or increased severity of illness among PLHIV, interruption in access to HIV medication and services, or any correlation between COVID-19 and HIV/AIDS” seems to be a sweeping statement or generalization. The authors can consider either providing a citation to support this statement or tempering this statement to say that this is according to the authors’ knowledge, or that there are few studies that study this phenomenon. 

This has been revised to reflect the suggestion by the reviewer

Line 82-83

4

Regarding data collection, how long did the telephone-based interviews last? Was Also, were incentives provided for participants to participate?  In the case where there were no monetary incentives, were there other considerations to acknowledge people’s time?

The telephone-based interviews took an average of thirty-five minutes. No incentives were given to participants. However, participants were advised to select convenient time for the interviews to be conducted

Line 163-165

5

Being that data was collected at three points (July, August, and October 2020), was this the same survey administered to three different groupsof people? Was the retrospective review used to supplement the survey?  When is the retrospective review factored into the analysis?  The methodology is not quite clear.

Being a prospective study, the survey was administered to the same cohort at the three points. This component targeted participants in Kibera informal settlement. The retrospective review of data targeted 7 hotspot counties to establish the changes in access to HIV Care and treatment services before and during the first wave of COVID 19. This has been clarified in the methodology section

Line  101-174

6

Overall, this is a pertinent, important, and timely topic.  The methods of the paper are a bit unclear.  Attending to these clarifying questions may help to improve the paper.

The comment is well noted and the methods revised accordingly

Line 101-174

Kind Regards,

Yvonne Opanga

Round 2

Reviewer 1 Report

My questions have been taken into account by the authors. Their answers are satisfactory. 

Author Response

The comments are well noted and appreciated.

Reviewer 3 Report

The authors have done well in responding to the suggested feedback.  The revised paper is clearer and appears suitable for publication.

Author Response

(The authors gave the same response as above.)
